# Evolutionary Dynamics of *FLC*-like MADS-Box Genes in Brassicaceae

**DOI:** 10.3390/plants12183281

**Published:** 2023-09-15

**Authors:** Lydia Gramzow, Renu Sharma, Günter Theißen

**Affiliations:** Matthias Schleiden Institute/Genetics, Friedrich Schiller University Jena, 07743 Jena, Germany

**Keywords:** MADS-box gene, phylogenomics, comparative transcriptomics, transcription factor

## Abstract

MADS-box genes encode transcription factors that play important roles in the development and evolution of plants. There are more than a dozen clades of MADS-box genes in angiosperms, of which those with functions in the specification of floral organ identity are especially well-known. From what has been elucidated in the model plant *Arabidopsis thaliana*, the clade of *FLC*-like MADS-box genes, comprising *FLC*-like genes *sensu strictu* and *MAF*-like genes, are somewhat special among the MADS-box genes of plants since *FLC*-like genes, especially *MAF*-like genes, show unusual evolutionary dynamics, in that they generate clusters of tandemly duplicated genes. Here, we make use of the latest genomic data of Brassicaceae to study this remarkable feature of the *FLC*-like genes in a phylogenetic context. We have identified all *FLC*-like genes in the genomes of 29 species of Brassicaceae and reconstructed the phylogeny of these genes employing a Maximum Likelihood method. In addition, we conducted selection analyses using PAML. Our results reveal that there are three major clades of *FLC*-like genes in Brassicaceae that all evolve under purifying selection but with remarkably different strengths. We confirm that the tandem arrangement of *MAF*-like genes in the genomes of Brassicaceae resulted in a high rate of duplications and losses. Interestingly, *MAF*-like genes also seem to be prone to transposition. Considering the role of *FLC*-like genes *sensu lato* (*s.l.*) in the timing of floral transition, we hypothesize that this rapid evolution of the *MAF*-like genes was a main contributor to the successful adaptation of Brassicaceae to different environments.

## 1. Introduction

MADS-box genes comprise a pan-eukaryotic gene family that is well-known for its involvement in nearly all aspects of plant development [1,2]. MADS-box genes encode for transcription factors that share a highly conserved DNA-binding domain, the MADS domain. The MADS domain is named after four of the first five proteins found to harbor this domain: MINICHROMOSOME MAINTENANCE1 (MCM1) from baker’s yeast (*Saccharomyces cerevisiae*), AGAMOUS (AG) from thale cress (*Arabidopsis thaliana*), DEFICIENS (DEF) from snapdragon (*Antirrhinum majus*), and SERUM RESPONSE FACTOR (SRF) from human (*Homo sapiens*) [3]. Two types of MADS-box genes can be distinguished, Type I and Type II. Most Type II genes of plants encode a particular domain structure where the MADS (M-) domain is followed by an Intervening (I-), a Keratin-like (K-), and a C-terminal (C-) domain; hence, these genes are also called MIKC-type MADS-box genes [4]. Several clades of MIKC-type genes had been established before the diversification of flowering plants [5,6]. One of these clades is the clade of *FLOWERING LOCUS C* (*FLC*)-like genes. The *FLC*-like genes are rather unusual among the MIKC-type MADS-box genes in that they appear to be lost or tandemly duplicated more frequently than most other clades of MIKC-type genes [7,8]. For example, *FLC*-like genes seem to be absent from the genomes of legumes, *Carica papaya*, *Citrus sinensis*, *Solanum lycopersicum*, and *Nelumbo nucifera* [7,9,10]. A number of tandem duplications of *FLC*-like genes have been observed in the Brassicaceae (crucifer) family [8].

*FLC* of the model plant *Arabidopsis thaliana* is a very well-studied gene. *FLC* has been identified as the causative locus in early flowering mutants, indicating that its gene product represses flowering [11]. In fact, the FLC protein binds to hundreds of target genes, among them important genes involved in the control of flowering, such as *FLOWERING LOCUS T* (*FT*), *SQUAMOSA PROMOTER BINDING PROTEIN-LIKE 15* (*SPL15*), *SUPPRESSOR OF OVEREXPRESSION OF CONSTANS 1* (*SOC1*), and *SEPALLATA 3* (*SEP3*) [12,13,14].

A variety of environmental and developmental cues, like photoperiod and age, modulate flowering time [15]. Plants growing in temperate climates often also require a prolonged period of cold before flowering is induced, a process called vernalization. *FLC* plays a central role in vernalization in a number of *A. thaliana* accessions [11,16]. The expression of *FLC* decreases with exposure to cold, the (epi)genetic control of which has been studied in quite some detail [17].

The repression of *FLC* expression has been sectioned into three phases. When plants are exposed to cold in the first phase, transcription of *FLC* is repressed by histone deacetylation [18] and upregulation of the long non-coding RNA COOLAIR [19,20,21]. COOLAIR refers to a collection of antisense transcripts of the *FLC* locus that can be divided into three major classes [22,23,24]. One class is unspliced. In a second class, a proximal splice site and proximal polyadenylation sites located in the sixth intron of *FLC* are used. In another class, a distal splice site and distal polyadenylation sites corresponding to the promotor of *FLC* are used. Expression of COOLAIR using proximal splice and polyadenylation sites leads to repression of the expression of *FLC* by chromatin modification [24]. It must be noted, however, that COOLAIR is not necessary for vernalization [19].

In the second phase of vernalization, in a period of prolonged cold, *FLC* is epigenetically silenced [25,26]. Polycomb repressive complex 2 (PRC2) is recruited to a region of the *FLC* locus downstream of the transcriptional start site, the so-called nucleation region [27]. Recruitment of PRC2 leads to the accumulation of histone H3K27me3 marks and, consequently, to the repression of *FLC*. Recruitment of PRC2 may involve another long non-coding RNA, COLDAIR [27]. Finally, when the temperature is rising again, the repressing histone marks H3K27me3 are spread over the whole locus. This way, a stable, heritable silencing of the *FLC* locus is established [28].

There are five genes that are closely related to *FLC* in *A. thaliana*: *MADS AFFECTING FLOWERING* (*MAF*) *1* to *MAF5* [29]. We will refer to the clade of *FLC*- and *MAF*-like genes together as *FLC*-like genes *sensu lato* (*s.l.*). Genes most closely related to *FLC* and more distantly to *MAF*-like genes, i.e., putative orthologs of *FLC* from *A. thaliana*, we will refer to as *FLC*-like genes *sensu strictu* (*s.str*.). As their name suggests, the *MAF*-like genes are also involved in the regulation of flowering time [30,31,32]. *MAF1*, also known as *FLOWERING LOCUS M* (*FLM*), regulates flowering in response to ambient temperature [33,34] and photoperiod [35]. *MAF2* has been shown to prevent precocious flowering in response to short periods of cold [31].

*MAF2* to *MAF5* occur in a cluster on chromosome 5, which seems to have expanded by tandem duplications. It has been hypothesized that all *MAF*-like genes in *A. thaliana* originated from a single ancestral gene in the common ancestor of Brassicaceae [8]. A first tandem duplication gave rise to two *MAF*-like genes in the ancestor of core Brassicaceae. Subsequently, independent tandem duplications, as well as a transposition, led to the cluster of *MAF*-like genes in *A. thaliana*. The tandem arrangement makes *MAF*-like genes susceptible to mutations leading to copy number variations (CNVs), which occur at a high rate [36]. Partial and complete duplications have been observed in different accessions of *A. thaliana* [37,38], and the number of *MAF*-like genes differs between different species of Brassicaceae [8,39]. It has been hypothesized that the rapidly evolving cluster of *MAF*-like genes facilitates swift adaptation of *A. thaliana* and potentially other Brassicaceae to changes in ambient temperature [8].

Brassicaceae (Cruciferae, cabbage, or mustard family) is a scientifically and economically important family of flowering plants. It contains the model plant *A. thaliana,* as well as significant crops like broccoli, turnip, radish, and rapeseed [40]. *Aethionemeae*, including *Aethionema arabicum,* represents the earliest branching tribe of Brassicaceae. The remaining tribes, termed core Brassicaceae, have been classified into three or five lineages based on phylogeny reconstructions [40,41,42]. Here, we adopt the system of three lineages according to [40] (Figure 1 and Figure 2). To date, for each of the lineages I and II, the genomes of more than ten species have been sequenced, including *A. thaliana*, *Capsella rubella*, *Lepidium sativum* (lineage I) and *Brassica rapa*, *Eutrema salsugineum*, *Arabis alpina* (lineage II) [43,44,45,46,47] (Brassicales Map Alignment Project, DOE-JGI, http://bmap.jgi.doe.gov/ (last accessed 8 September 2023)). In contrast, the genomes of only a few species of lineage III, like *Euclidium syriacum* and *Diptychocarpus strictus,* have been sequenced. The genome of *Ae. arabicum* has also been sequenced [48].

Most MIKC-type MADS-box genes of plants reveal a characteristic, conservative mode of evolution characterized by the preferential retention and the sub-, and neofunctionalization of copies after whole genome duplications, but relatively few gene losses and single and tandem gene duplications [49]. However, a previous analysis suggested that the evolutionary dynamics of *FLC*-like genes are quite different even though they also represent MIKC-type genes [8]. This conclusion was based on a strictly limited amount of data, however. To test our hypothesis that the evolution of *FLC*-like genes is somewhat different from the other MIKC-type MADS-box genes, we here make use of the increasing availability of genome and transcriptome data for Brassicaceae to study the evolutionary dynamics of *FLC*-like genes *s.l.* in more depth and detail. Our study is, to the best of our knowledge, the first comprehensive investigation of *FLC*-like genes in the evolutionary, ecologically, and agronomically important plant family of Brassicaceae on the basis of whole genome sequence data. We find that there are three major clades of *FLC*-like genes *s.l.,* which all evolve under purifying selection but with substantially different strengths. We show that not only in *A. thaliana* but potentially in all Brassicaceae, the *MAF* cluster undergoes rapid changes, which may have contributed to the success of the mustard family in adapting to different environments in approximately the past 35 million years.

**Figure 1 plants-12-03281-f001:**
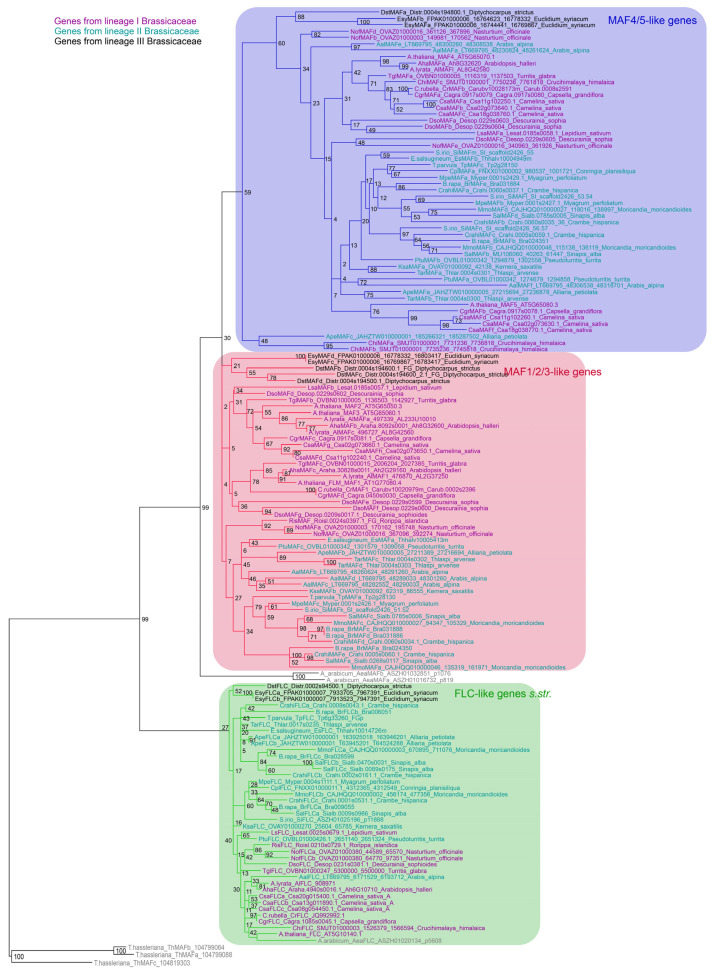
Phylogeny of *FLC*-like genes *s.l.* The phylogeny was reconstructed using RaxML [50] based on a protein alignment of conceptually translated coding sequences. Numbers on the nodes represent bootstrap values. Gene names are colored according to the lineage of Brassicaceae in which they were identified. The clades are shaded: blue, *MAF4/5*-like genes; red, *MAF1/2/3*-like genes; green, *FLC*-like genes *s.str. FLC*-like genes from *Tarenaya hassleriana* were used as sister and outgroup representatives.

**Figure 2 plants-12-03281-f002:**
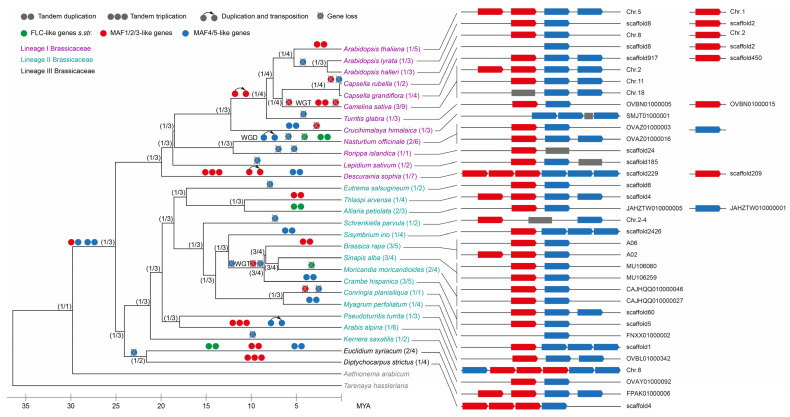
Phylogeny of Brassicaceae species and *Tarenaya hassleriana* showing events in the evolution of *FLC*-like genes *s.l.* and genomic arrangement of *MAF*-like genes. This phylogeny is based on the one by Walden et al., 2020 [40] and was pruned to include only the species studied here. Species names are colored according to the lineage of Brassicaceae they belong to, as explained at the top left. The numbers of identified *FLC*-like genes *s.str.* and *MAF*-like genes are indicated for each species in brackets after the species name. Based on the phylogeny of *FLC*-like genes *s.l.* (Figure 1), duplications, triplications, duplications involving transposition, and gene losses were inferred for *FLC*-like genes *s.str.*, *MAF1/2/3*- and *MAF4/5*-like genes and are marked on the phylogeny. The meaning of the symbols is explained above the phylogeny. If there are several events on one branch, the order and the timing of these events are unknown and were chosen arbitrarily. The genomic arrangement of the *MAF*-like genes is shown on the right, where *MAF1/2/3*-like genes are indicated by red boxes and *MAF4/5*-like genes are shown as blue boxes. The size of the genes and the distance between the genes are not drawn to scale. Gray boxes represent incomplete genes.

## 2. Results

### 2.1. Phylogeny and Selection Pressure of FLC-like Genes s.l. in Brassicaceae

To investigate the evolutionary dynamics of *FLC*-like genes, we collected or annotated these genes from the genomes of 29 Brassicaceae species (Appendix A). We identified at least two *FLC*-like genes *s.l.* in all species examined (Figure 1 and Figure 2). Most *FLC*-like genes *s.l.* have seven coding exons (Appendix A). The lengths of the genomic loci vary between around 2 kilo base pairs (kbp) to more than 15 kbp. Apart from a few exceptions, the MADS domain is encoded by the first exon. In the cases where the exon-intron structure deviates from the general pattern of seven exons and the complete MADS domain encoded by the first coding exon, it remains to be shown as to whether these loci represent functional genes.

To separate *FLC*-like genes *s.str.* and *MAF*-like genes, we reconstructed the phylogeny of these genes (Figure 1). Indeed, *FLC*-like genes *s.str.* and *MAF*-like genes form two clearly separable clades in our phylogeny. All species contain at least one *FLC*-like gene *s.str.,* where the genomes of eight species encode more than one *FLC*-like gene *s.str.*: *Nasturtium officinale*, *Alliaria petiolata*, *Moricandia moricandioides,* and *Euclidium syriacum* have two *FLC*-like genes; *Camelina sativa*, *Brassica rapa*, *Sinapis alba,* and *Crambe hispanica* possess three *FLC*-like genes *s.str.* (Figure 1 and Figure 2). Similarly, all species encode at least one *MAF*-like gene. However, the number of *MAF*-like genes is more variable, ranging from one to nine per species. We also found loci potentially harboring partial *MAF*-like genes possibly representing pseudogenes (Figure 2). However, as it is often difficult to distinguish pseudogenes from unusual functional genes, we did not investigate this issue further. In any case, our phylogeny suggests that the ancestor of Brassicaceae likely possessed one *FLC*- and one *MAF*-like gene (Figure 2), as has been hypothesized previously [8].

Furthermore, our phylogeny confirms the presence of two clades of *MAF*-like genes in core Brassicaceae (Figure 1), as already proposed [8]. This suggests that an ancestral *MAF*-like gene has been duplicated in an ancestor of core Brassicaceae. It is not clear, however, if this duplication happened before or after *Ae. arabicum* branched off as there is a sequencing gap in the genomic locus of the *MAF* cluster in *Ae. arabicum* [48]. One of these clades contains *MAF1*, *MAF2,* and *MAF3* from *A. thaliana* while the other includes *MAF4* and *MAF5*. Therefore, the clades were termed *MAF1/2/3* and *MAF4/5* (Figure 1). We also analyzed the genomic location of the *MAF*-like genes. Mostly, genes from both of the clades are arranged in tandem clusters (Figure 2). Consequently, the duplication giving rise to the *MAF1/2/3* and *MAF4/5* clades was almost certainly a tandem duplication (Figure 2).

Interestingly, the branch lengths of the *FLC*-like genes *s.str*., the *MAF1/2/3*-like genes, and the *MAF4/5*-like genes appear quite different in our phylogeny: While *FLC*-like genes *s.str*. have relatively short branches, *MAF1/2/3*-like genes have branches of intermediate lengths and the *MAF4/5*-like genes have comparably long branches (Figure 1). To substantiate this impression of different rates of evolution, we conducted selection analyses and determined the ratio of non-synonymous substitution rate (dN) to synonymous substitution rate (dS), ω. A branch model allowing different values for ω for the three clades offered a significantly better fit than a model assuming the same ω ratio for all genes (Table 1). According to the branch model Mb, all three clades are evolving under purifying selection with ω ratios of less than one. However, the ω ratio is nearly double for *MAF4/5*-like genes at 0.41, as for *FLC*-like genes at 0.21, while the ω ratio of *MAF1/2/3*-like genes is intermediate at 0.35. This finding suggests that much stronger purifying selection acted on *FLC*-like genes *s.str.* than on *MAF1/2/3*- and *MAF4/5*-like genes.

### 2.2. Duplications and Losses of FLC-like Genes s.l.

Next, we analyzed duplications and losses of *FLC*-like genes *s.l.* (Figure 2). The two *FLC*-like genes *s.str.* of *Nasturtium officinale* are about 12 kilo base pairs (kbp) apart and are likely the result of a tandem duplication. Interestingly, based on the fact that the whole *MAF* cluster is duplicated in *N. officinale*, we hypothesize that there was a recent whole genome duplication (WGD) in this lineage that is probably shared with the closely related *Cardamine amara* [51] and *Cardamine enshiensis* [52]. However, no *FLC* duplicate resulting from this WGD, that would be expected to be located on a different scaffold, was identified. The MADS-boxes of the two *FLC*-like genes *s.str.* of *Euclidium syriacum* are about 13 kbp apart. However, the two genes are facing each other and are likely the result of an inverted duplication. In *Alliaria petiolata*, the two *FLC*-like genes are more than 500 kbp apart but still on the same scaffold and probably originated from a segmental duplication. *Camelina sativa,* as well as the Brassiceae *Brassica rapa*, *Sinapis alba*, *Moricandia moricandioides,* and *Crambe hispanica*, experienced a whole genome triplication (WGT) in the lineages leading to them [53,54]. Consequently, all of these species hold three *FLC*-like genes *s.str*. except *M. moricandioides*, which has only two *FLC*-like genes *s.str*. and must have hence lost one of these genes. However, as mentioned above, there is no species without an *FLC*-like genes *s.str.*

More gene duplications and losses were observed in the *MAF*-like genes (Figure 2). First, a tandem duplication in an ancestor of core Brassicaceae gave rise to the *MAF1/2/3* and *MAF4/5* clades, as mentioned above. After the WGT in *C. sativa* and the WGD *N. officinale*, the three and two resulting *MAF*-loci, respectively, were partially conserved (Figure 2). In contrast, one of the three *MAF* clusters originating from the WGT in Brassiceae was apparently lost quickly after the triplication, as *B. rapa*, *S. alba*, *M. moricandioides,* and *C. hispanica* all have only two loci left.

In the clade of *MAF1/2/3*-like genes, we observe five and three tandem duplications and triplications, respectively (Figure 2). Furthermore, we assume that *MAF1/2/3*-like genes have been duplicated twice involving transposition and were lost six times independently. Nevertheless, *Conringia planisiliqua* is the only species among the ones we examined that has no *MAF1/2/3*-like genes at all.

For the *MAF4/5*-like genes, these events are not easily deducible from the phylogeny as the accelerated rate of evolution results in a phylogeny that deviates to a large extent from the species phylogeny. Based on our phylogeny, the genomic arrangement of the genes and parsimony considerations, we assume that there was an early tandem duplication of *MAF4/5*-like genes in an ancestor of Brassicaceae after the lineage that led to *Ae. arabicum* had branched off (Figure 2). In this case, we infer a total of seven tandem duplications, as well as two duplications involving transposition in the *MAF4/5* clade. Additionally, this scenario involves 13 losses of *MAF4/5*-like genes. Similar to the situation for the *MAF1/2/3*-like genes, there is only one species, *Rorippa islandica*, that no longer encodes an *MAF4/5*-like gene.

Hence, the number of duplication and loss events in the evolution of the different clades of *FLC*-like genes *s.l.* reflects the selection pattern described above. For the clade of *FLC*-like gene *s.str.* which is under stronger purifying selection, the lowest number of duplication and loss events was observed, while this number was higher for the *MAF1/2/3*-like and the *MAF4/5*-like genes, which evolve under less purifying selection.

## 3. Discussion

Our study reveals, in line with previous studies [8], three major clades of *FLC*-like genes *s.l.* in Brassicaceae. The *FLC*-like genes *s.str.* were identified in all of the species analyzed and found to evolve under strong purifying selection. The ω ratio of 0.21 that we observed for the *FLC*-like genes *s.str.* is similar to the one that was observed for *FLC*-like genes in *Brassica rapa* [55]. In contrast, in *Brassica napus* and *Boechera stricta* individual *FLC*-like genes were shown to evolve neutrally with ω ratio of around one [56,57]. However, neutral evolution is only observed when more than one *FLC*-like gene *s.str.* is present in a species and thus may reflect functional redundancy of the specific genes, but it does not reflect the general evolution pattern in Brassicaceae.

Only a few duplication events were inferred for the *FLC*-like gene *s.str.* These were mainly observed after polyploidization events, such as in *Sinapis alba*, *Crambe hispanica,* and *B. rapa* (this study) or *Brassica oleracea* and *B. napus* [58,59]. *B. oleraceae* contains three *FLC*-like genes *s.str.* while the genome of *B. napus* even carries nine of these genes [58,59]. The low number of duplications, apart from those caused by polyploidization events, indicates that the function of these genes is sensitive to gene dosage. Furthermore, the relatively strong purifying selection suggests that *FLC*-like gene *s.str.* have essential functions in Brassicaceae. *FLC* of *A. thaliana* (*AtFLC*) is well known for its role in vernalization and floral transition [11,16]. Expression of *AtFLC* decreases with extended exposure to cold, i.e., during vernalization. Other functions of *AtFLC* include temperature-dependent regulation of germination [60] and temperature compensation of the circadian clock [61]. In other Brassicaceae species, *FLC*-like genes *s.str.* have also been revealed to be important for the vernalization response. Mutations of *FLC*-like genes have been shown to reduce the requirement for vernalization in *Capsella rubella* [62], *Camelina sativa* [53], *Thlaspi arvense* [63], *Arabis alpina* [64], and *Cardamine occulta* (a species that we did not study here) [65]. Furthermore, expression of *FLC*-like genes *s.str.* decreases with prolonged vernalization treatment in *A. lyrata* [66], *A. halleri* [67], *B. rapa* [68], and *S. alba* [69], indicating a similar function of *FLC*-like genes *s.str*. in these species, as well. However, it is quite likely that not all of the Brassicaceae species require vernalization, like *Nasturtium officinale* [70], as well as *C. hispanica* and *Euclidium syriacum*, the latter two species mainly growing in the subtropical biome [71]. Furthermore, it has been hypothesized that Brassicaceae originated as a tropical/subtropical family in a warm and humid climate [72]. Subsequently, during an episode of global cooling, Brassicaceae adapted to dryer and cooler climates. Hence, while involvement of *FLC*-like genes *s.str*. in vernalization responses has thus been shown for a diversity of Brassicaceae species, this finding likely does not reflect the ancestral function and does not completely explain the strong conservation of *FLC*-like genes *s.str.* in Brassicaceae. Reports of other potential functions of *FLC*-like genes *s.str*. remain scarce. In *Cardamine hirsuta*, the *FLC*-like gene *s.str*. affects leaf form, where accessions with low expression of *ChFLC* develop fewer leaflets than accessions with higher expression of *ChFLC* [73]. In *Ae. arabicum*, the *FLC*-like gene *s.str.* has been identified by QTL analysis as a candidate for a major regulator of glucosinolate content [74]. Glucosinolates are important metabolites involved in plant defense in Brassicaceae [75]. Based on this limited additional functional information, we suggest that the ancestral function of *FLC*-like genes *s.str.* was more general in the determination of flowering in response to environmental cues and probably also in the coordination of different aspects of development. Parts of this ancestral function are likely still conserved in most Brassicaceae species resulting in the strong selection pressure that was found to act on *FLC*-like genes *s.str.*

Outside of Brassicaceae, *FLC*-like genes *s.l.*, i.e., *FLC*-like genes that are equally related to *MAF*-like genes and *FLC*-like genes *s.str.* of the Brassicaceae, have also been characterized. *Citrus clementina* belongs to the order Sapindales, which diverged from Brassicales about 105 million years ago (MYA). The expression of the *FLC*-like gene of this species correlates with the inhibition of flowering in plants with developing fruits [76]. For *Punica granatum* belonging to the order Myrtales that diverged from Brassicales about 110 MYA), it has been found that deleterious mutations of an *FLC*-like gene result in evergreen rather than deciduous trees [77]. In *Malus domestica*, belonging to Rosales, of which the most recent common ancestor (MRCA) with Brassicales existed about 120 MYA, expression of *FLC*-like genes *s.l.* has been shown to decrease drastically during dormancy when growth is ceased during winter [78]. In *Beta vulgaris* belonging to Caryophyllales and *Cichorium indybus* belonging to Asterales, both of which separated from Brassicales about 125 MYA, expression of the *FLC*-like genes *s.l.* decrease in response to cold [79,80]. However, at least for *B. vulgaris,* it has been revealed that the *FLC*-like gene *s.l.* does not represent the gene responsible for the vernalization requirement [81]. In contrast, in *Carthamus tinctorius,* also belonging to Asterales, as well as in *Actinidia chinensis,* belonging to Ericales, both having the same evolutionary distance to Brassicales, expression of *FLC*-like genes is induced by cold [82,83]. In *Actinidia chinensis*, cold-responsiveness of the *FLC*-like gene has additionally been shown to control budbreak. Grasses belonging to the order Poales diverged from Brassicales about 160 MYA. In several grass species, including *Triticum aestivum*, *Hordeum vulgare,* and *Brachypodium distachyon*, expression changes of *FLC*-like genes *s.l.* have also been tightly linked to vernalization [84,85,86]. Hence, broadly speaking, *FLC*-like genes are mostly involved in sensing cold and integrating development with this environmental cue in species outside Brassicaceae. Temperature-dependent regulation of development, likely not restricted to vernalization, may thus represent the ancestral role of *FLC*-like genes *s.l.* and explain the strong purifying selection on these genes in Brassicaceae, at least in part.

In contrast to the *FLC*-like genes *s.str*., genes of the two clades of *MAF*-like genes which probably originated in the stem group of core Brassicaceae after *Ae. arabicum* branched off, have been duplicated and lost several times in Brassicaceae and experience less purifying selection (Figure 2). The five *MAF*-like genes of *A. thaliana* are partially redundant and are involved in floral repression [32]. Similarly, the *MAF*-like genes of *A. alpina* act as repressors of flowering [39]. The facts that at least one *MAF*-like gene was identified from every species analyzed and the still purifying selection acting on both clades of *MAF*-like genes, though at a significantly lower strength, suggest that retention of *MAF*-like genes is of selective advantage for Brassicaceae species. It has been suggested that the selective advantage may be conferring alterations in flowering time in response to environmental change [8].

We observed a number of duplications of *MAF*-like genes, including four duplications involving transposition (Figure 2). The duplication involving transposition giving rise to *FLM* has been observed previously [8]. We can confine the time of transposition to an ancestor of the tribe Camelineae after Turritideae branched off. Duplication involving transposition was also observed before for *MAF*-like genes of *Arabis montbretiana* [39]. Taken together, these findings emphasize the role of independent transpositions of *MAF*-like genes in different Brassicaceae lineages. Additionally, *MAF*-like genes propagated by tandem duplications and triplications. In total, we observed 14 of these events (Figure 2). It has been observed that genes that expanded via tandem duplications (and triplications) are frequently involved in responses to environmental stimuli [87], a finding which very likely holds true for the *MAF*-like genes. The fact that we also observe 16 gene losses within the clade of *MAF*-like genes, however, also demonstrates that not all newly arising *MAF*-like genes are of adaptive value, at least not in the long run.

### Conclusions and Future Perspectives

*FLC*-like genes *s.l.* have essential functions in Brassicaceae in temperature-dependent regulation of development and flowering time. Our analyses (this work and Ref. [8]) confirm that the number of *MAF*-like genes rapidly evolved by small-scale duplications and losses, suggesting that this variability in gene number facilitated adaptation to changes in temperature, a finding which is of special interest in a world of climate change. Overall, the *FLC*-like genes *s.l.* of Brassicaceae reveal a fascinating diversity in evolutionary dynamics, ranging from the relatively highly conserved single-copy *FLC*-like genes *s.str.* to the much more dynamic *MAF4/5*-like genes. Our work confirms previous reports based on more limited data [8] that concluded that the tandem arrangement of *MAF*-like genes in the genomes of Brassicaceae resulted in a high rate of duplications and losses. Interestingly, our analyses suggest that *MAF*-like genes are prone to transposition. Considering the role of *FLC*-like genes *s.l.* in the timing of floral transition, we hypothesize that this rapid evolution of the *MAF*-like genes was a major contributor to the successful adaptation of Brassicaceae to different environments. Significant future work will be required to test that hypothesis and to determine all the functional implications of the evolutionary dynamics of *FLC*-like genes *s.l.* At first, functions of more *FLC*-like genes in phylogenetic informative taxa need to be determined, e.g., by reverse genetics employing CRISPR-Cas. Ideally, the phenotype of mutants would be compared to wild-type plants under natural growth conditions. Mapping the different functions of *FLC*-like genes on a tree of Brassicaceae will enable the identification of ancestral and derived states.

In any case, *FLC*-like genes *s.l.* in Brassicaceae represent an attractive model system to investigate the molecular evolution and functional implications of closely related paralogous genes in a well-defined and intensively studied plant family.

## 4. Materials and Methods

### 4.1. Identification of FLC-like Genes

For the *FLC*-like genes, we used the dataset of [8] as a starting point. Additional *FLC*-like genes were identified on different databases, as summarized in Appendix A. For the genomes assessed on the Brassicaceae Database (BRAD) [88], we used the option to search “Syntenic Gene @ Subgenomes” with AT5G10140 (*FLC*), AT1G77080 (*MAF1/FLM*), and AT5G65070 (*MAF4*) as queries. To ensure some potentially transposed *FLC*-like genes were not overlooked, we also conducted blastn searches [89] using the sequences of *FLC* and *MAF4* as queries. For the species assessed on Phytozome [90] and NCBI genomes [91], we conducted blastn searches with the same query sequences. In case a gene was already annotated at the resulting locus on Phytozome, we saved the coding sequences of that gene. For BLAST hits on NCBI genomes and if no gene was annotated on Phytozome, we tried to predict a gene at the resulting locus using FGENESH+ [92] with the protein sequences of either FLC or MAF4 as a guide and “Dicot plants, Arabidopsis (generic)” as organism. Subsequently, we saved the coding sequence of the predicted gene. The amino acid sequences are given in Appendix A. Using the genomic and coding sequences, we determined the exon-intron structure with the webserver “Gene structure display server” [93].

### 4.2. Phylogeny Reconstruction

For phylogeny reconstruction, we conceptionally translated the coding sequences of the identified *FLC*-like genes *s.l.* into protein sequences using EMBOSS Transeq [94]. The protein sequences were aligned using MAFFT [95] on the CIPRES Science gateway [96]. The alignment was manually trimmed on Jalview [97] to remove regions with little conservation. Based on the trimmed alignment, the phylogeny was reconstructed using RAxML [50] as offered on the CIPRES Science gateway [96]. We used the JTT substitution model [98] with invariable sites and a discrete gamma model for variation among sites. We reconstructed 1000 bootstrap values.

Based on our phylogeny, we deduced a hypothesis on how many *FLC*-like genes *s.l.* were present in the most recent common ancestors (MRCAs) of different species (Figure 2). If not as many *FLC*-like genes *s.l.* were found as expected by the reconstruction of the gene content in the MRCAs, we assumed that *FLC*-like genes *s.l.* must have been lost.

### 4.3. Selection Analyses

For the selection analyses, the un-trimmed MAFFT-alignment was reverse translated into a codon alignment using RevTrans-2.0 [99]. The resulting codon alignment was trimmed in the same way that we trimmed the protein alignment previously. Furthermore, branch-length information from the phylogeny obtained by using RAxML was removed. The trimmed nucleotide alignment and the phylogeny were subsequently converted into files readable by the PAML program [100]. Using codeml of the PAML program, we estimated parameters of sequence evolution for a null-model assuming the same ratio of nonsynonymous to synonymous substitution rates (ω = dN/dS) for all branches in the phylogeny and for a branch-model allowing different ω ratios for branches leading to *FLC*-like genes *s.str*., *MAF1/2/3*-like genes and *MAF4/5*-like genes. Analyses were started three times for each model, with initial ω ratios of 0.5, 1, and 2, respectively, to check the convergence of the analyses. The resulting log Likelihood (lnL) values were compared using a likelihood ratio test (LRT) to infer which model fits our data best.

## Figures and Tables

**Table 1 plants-12-03281-t001:** Selection analyses of *FLC*-like genes *s.l.* Model M0 denotes the null-model assuming the same ratio of nonsynonymous to synonymous substitution rates (ω) for all branches in the phylogeny. The branch-model Mb was defined as allowing different ω ratios for branches leading to *FLC*-like genes *s.str.*, *MAF1/2/3*-like genes, and *MAF4/5*-like genes. ω0 represents ω for all branches in case of M0 and ω for all branches not belonging to the *FLC*-like genes *s.str*.; *MAF1/2/3*-like genes and *MAF4/5*-like genes in case of Mb; ω1 represents ω for *MAF4/5*-like genes; ω2 represents ω for *MAF1/2/3*-like genes; ω3 represents ω for *FLC*-like genes *s.str.*; lnL, log likelihood; np, number of parameters; κ, transition/transversion rate; 2ΔLnL, twice the difference of the log likelihood values; Δnp, difference in the number of parameters; LRT, *p*-value of likelihood ratio test of Mb model relative to the null M0 model.

Results of Selection Models								Likelihood Ratio Test of Mb vs. M0		
Model	lnL	np	ω0	ω1	ω2	ω3	κ	2ΔLnL	Δnp	LRT
M0	−19,989.78	292	0.34	-	-	-	1.99	24.03	3	0.000025
Mb	−19,965.76	295	0.33	0.41	0.35	0.21	1.99			

## Data Availability

The data presented in this study are available in the Appendix A.

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
