# Peer review of "Evolutionary Dynamics of FLC-like MADS-Box Genes in Brassicaceae"

_plants, 2023, doi:10.3390/plants12183281_

Round 1
Reviewer 1 Report
The data presented by authors are not sufficient to support one good publication in the current status, althought the current data is nicely provided. The anaysis about B.oleracea and B. napus should be included at least.
Reviewer 2 Report
The present work dealing with Evolutionary dynamics of FLC-like MADS-box genes in Brassicaceae presented several insights of the FLC-like MADS-box genes in Brassicaceae which are useful to understand the adaptation mechanism of plants. But there are some improvement required which must be addressed.
The abstract contains long background and some general results.
Abstract must contain methods in brief and specific findings.
Conclusion and future recommendations should also be added.
How the findings justify the hypothesis of the study should be added in the abstract.
Paragraph 97 -109 should add some information such as the presence of FLC-like MADS-box genes in brassica species, its function and role and evolutionary dynamics.
Discuss novelty of the study in introduction.
Add justification for the study, and a clear research question or hypothesis
Section 4.1 and 4.2 could be cited with relevant studies https://doi.org/10.1016/j.plaphy.2021.01.042
The study should include gene structure and function analysis it would be better to provide thorough understanding of the gene family.
The branch length of the phylogeny analysis is very different and do not support the objective of the study.
Line 191-194 provide further reasons in discussion and justify with the light of relevant studies.
How the authors determined losses events of the genes should be added in methods or results.
Add conclusion, future perspective and research gap of the study.
Long sentences should be avoided to convey clear meaning to readers
Reviewer 3 Report
The subject of the research presented in the manuscript entitled “Evolutionary dynamics of FLC-like MADS-box genes in Brassicaceae” written by Lydia Gramzow, Renu Sharma and Günter Theißen is connected to the activity of FLC like MADS-box genes. The plant family chosen is commonly known, its representatives are crop plants that provide primarily food or products for food production. Moreover, model organism Arabidopis thaliana or crop plant Brassica napus are frequently chosen as an object of research. On the other hand, MADS-box genes are commonly occurring genes that are detected in nearly all eukaryotes studied. The functions of this gene group are also well known and extensive.
As the authors emphasize at the end of the introduction, the aim of the research was to make use of the increasing availability of genome and transcriptome data for Brassicaceae to study the evolutionary dynamics of FLC-like genes s.l.
In the relatively long chapter “1. Introduction” of the reviewed paper, the authors described the MADS-box genes, characterized them, and emphasized their function. The analysis of these genes is the main part of the introduction. At the end of this part of the paper, the authors gave some data about the Brassicaceae family. The introduction contains content that is appropriate for the investigation undertaken by the authors.
In the second part of the reviewed publication, entitled “2. Results” the data obtained are described in detail. The documentation consists of 2 figures and 1 table. The figures are very bright and clear, and elaborate on the results of research. Undoubtedly, the authors conducted the research adequately to the planned aim of the study and showed it this chapter properly.
The next part, “3. Discussion” is large, and, as typical for the scientific papers, the results obtained by the authors are analyzed. This analysis is accurate and sufficiently detailed and adequate to the obtained results. The large amount of literature cited in this chapter of the paper is well-chosen.
“Part 4. Material and Methods,” is the short part of the paper and contains the description of 3 methods used by the authors of the reviewed publication. These methods are properly chosen and used. I did not notice any mistakes in the methods.
The last part of the work, i.e. “Literature,” is very extensive, and consists of as many as 95 items.
Going back to the abstract of this work, I find it proper and it compactly shows the research achievements of the authors presented in the reviewed paper.
Summary of the review:
I find this paper good and I think that it is suitable for publishing in “Plants”.
Reviewer 4 Report
This paper is purely bioinformatics analysis without some other necessary molecular biological verification means, and is not recommended for acceptance for publication.
Round 2
Reviewer 1 Report
Thanks for your positive reponse, we are agreeing to this possible publication in Plants
Author Response
Our reply: We are very grateful for the reviewer’s kind view.
Reviewer 4 Report
1. In Fig1. The format on the picture is not uniform, such as A thaliana-FLC AT5G...,EsyMAFb FPAK.... Euclidium.. and Distr 0004s.... Dptych... formats coexist, theoretically maintaining one format.
2. Line 43 reference 8 is the same with Line 398 [8]?
Author Response
- In Fig1. The format on the picture is not uniform, such as A thaliana-FLC AT5G...,EsyMAFb FPAK.... Euclidium.. and Distr 0004s.... Dptych... formats coexist, theoretically maintaining one format.
Our reply: The lack of uniformity is partially due to the fact that we used published gene names whenever they were available. We agree with the reviewer, however, that the naming of the newly described FLC-like genes s.l. was not uniform either and have changed the naming accordingly.
- Line 43 reference 8 is the same with Line 398 [8]?
Our reply: Yes, the reference is the same. The reference reports a first study on FLC-like genes in Brassicaceae. Hence, we started using the data from this reference and referred to it in line 398. In this reference, it was also found that FLC-like gene evolve a bit unusual as compared to most other MIKC-type genes. Hence, we also referred to this publication in line 43.